# Synergistic Effects and Kinetic Analysis in Co-Pyrolysis of Peanut Shells and Polypropylene

**DOI:** 10.3390/foods13081191

**Published:** 2024-04-13

**Authors:** Zhigang Huang, Jiahui Wu, Tenglun Yang, Zihan Wang, Tong Zhang, Fei Gao, Li Yang, Gang Li

**Affiliations:** 1School of Computer and Artificial Intelligence, Beijing Technology and Business University, Haidian District, Beijing 100048, China; huangzg@btbu.edu.cn (Z.H.); wjh99999909@163.com (J.W.); ytl1010@163.com (T.Y.); 19801298025@163.com (Z.W.); ztong000608@163.com (T.Z.); 2Beijing Key Laboratory of Quality Evaluation Technology for Hygiene and Safety of Plastics, Beijing Technology and Business University, No. 11 Fuchenglu, Haidian District, Beijing 100048, China; 3China-Canada Joint Lab of Food Nutrition and Health (Beijing), Key Laboratory of Special Food Supervision Technology for State Market Regulation, School of Food and Health, Beijing Technology and Business University, 11 Fucheng Road, Beijing 100048, China; feigao@btbu.edu.cn; 4School of International Studies, Peking University, Haidian District, Beijing 100871, China; lily_yang@pku.edu.cn

**Keywords:** co-pyrolysis process, thermogravimetry, heating rate, activation energy, synergistic mechanism

## Abstract

The impact of COVID-19 has boosted growth in the takeaway and medical industries but has also generated a large amount of plastic waste. Peanut shells (PS) are produced in large quantities and are challenging to recycle in China. Co-pyrolysis of peanut shells (PS) and polypropylene (PP) is an effective method for processing plastic waste and energy mitigation. Thermogravimetric analysis was conducted on PS, PP, and their blends (PS-PP) at different heating rates (10, 20, 30 °C·min^−1^). The results illustrated that the co-pyrolysis process of PS-PP was divided into two distinct decomposition stages. The first stage (170–400 °C) was predominantly linked to PS decomposition. The second stage (400–520 °C) resulted from the combinations of PS and PP’s thermal degradations, with the most contribution from PP degradation. With the increase in heating rate, thermogravimetric hysteresis appeared. Kinetic analysis indicated that the co-pyrolysis process reduced the individual pyrolysis activation energy, especially in the second stage, with a correlation coefficient (R^2^) generally maintained above 0.95. The multi-level reaction mechanism function model can effectively reveal the co-pyrolysis process mechanism. PS proved to be high-quality biomass for co-pyrolysis with PP, and all mixtures exhibited synergistic effects at a mixing ratio of 1:1 (PS1-PP1). This study accomplished effective waste utilization and optimized energy consumption. It holds significance in determining the interaction mechanism of mixed samples in the co-pyrolysis process.

## 1. Introduction

In recent years, with the increasing depletion of fossil energy and the growing threat of environmental pollution, global energy consumption is expected to increase by 28%, and a large amount of biomass and waste plastics cannot be recycled [1]. Biomass has limitations in terms of high oxygen content, low calorific value, and high transportation costs, which need to be further addressed [2]. In addition, the COVID-19 pandemic has caused much plastic pollution in takeaway, online shopping, and medical waste [3]. About 100 billion tons of plastic are discarded globally every year; 79% of plastic waste ends up in landfills or other environmental media, 13% is incinerated, and 9% is reused, of which polypropylene (PP) is the main component [4]. Improper landfilling and in situ incineration of waste accelerate fossil fuel depletion and contribute to the spread of the virus [5].

Nevertheless, biomass has the advantages of vast sources, large production, renewable properties, and CO_2_ neutralization, making it the fourth-largest energy system after coal, oil, and natural gas [6]. Oxygen-free radicals in biomass can promote the decomposition of plastics [7]. Besides, as a hydrogen-rich resource, plastic can be used to make up for the low hydrogen and high oxygen content of biomass fuels as a hydrogen donor, thus becoming an excellent auxiliary pyrolysis material [8]. From the recycling perspective, these two wastes have many similarities regarding chemical composition and thermochemical utilization [9]. It is expected to achieve complementary advantages through thermochemical conversion technology and provide an essential basis for the clean utilization of biomass resources.

Currently, pyrolysis technology can be divided into liquefaction, gasification, microwave pyrolysis, catalytic pyrolysis, and co-pyrolysis [10]. Among them, some scholars have found that co-pyrolysis technology can make full use of the advantages of various raw materials to achieve hydrogen transfer under ambient pressure without high-pressure hydrogenation [11]. Chen et al. (2020) studied the co-pyrolysis behavior of tobacco straw and PP and found that the organic gas production increased, the residual biochar decreased, and the reaction activation energy (Ea) decreased [12]. Yang et al. (2021) found that the inorganics present in the biomass improved the degradation of low-density polyethylene under fast co-pyrolysis at high temperatures, which promoted tar formation and suppressed the formation of water, gas, char, and coke [13]. The blend interaction affects the Ea and pyrolysis products and changes the entire kinetics, reaction mechanism, and operating conditions. Synergistic effects usually evaluate this interaction [14].

In addition, researchers have proposed several mechanisms to elucidate the synergistic effects of co-pyrolysis, including the hydrogen supply capacity of the plastic, the stability of the plastic, and the interaction of free radicals between volatiles [15]. Whereas, the interaction between biomass and PP in the respective pyrolysis process and the overall co-pyrolysis process still requires further investigation. The synergistic mechanism of the co-pyrolysis of biomass and PP has not been clearly explained [16]. Therefore, it is necessary to explore further the synergistic effect between the co-pyrolysis behavior and kinetics of biomass and PP to gain a deeper understanding of their co-pyrolysis process.

Consequently, the representative PS and PP as raw materials were selected to mainly study their co-pyrolysis characteristics and discuss the influence of synergistic effects on co-pyrolysis behavior during pyrolysis and the interaction mechanism. Firstly, a non-isothermal gravimetric analyzer was used to study the interaction between PP and PS in the co-pyrolysis process, as well as the influence of heating rate on the co-pyrolysis behavior. Then, the Coats–Redfern (CR) method carried out the kinetic analysis of different co-pyrolysis reaction stages, and the corresponding Ea was calculated. Finally, the physical structure of biochar was qualitatively analyzed by scanning electron microscopy.

## 2. Materials and Methods

### 2.1. Sample Preparation

PS was obtained from local farmers’ markets in Beijing; PP was derived from a garbage collection station in Fang Shan District (Beijing, China). PS and PP were dried in an oven at 120 °C for 10 h before experimenting. After that, PS and PP were crushed by a multi-functional pulverizer (XT-A400, Xintao, Jinhua, China) and sieved through a 50-mesh screen to achieve a consistent particle size of less than 250 µm. PS and PP were evenly mixed in different mass ratios (1:0, 1:1, 1:2, 2:1, 0:1). PS1-PP1 indicates that PS and PP are 1:1 mixed.

### 2.2. Sample Analysis

#### 2.2.1. Elemental Analysis

The main elemental compositions (C, H, N, and S) of the samples were determined by an elemental analyzer (Vario EL cube, Elementar, Frankfurt, Germany). Elemental analysis was performed using the X-ray Fluorescence (XRF) technique. The calibration curve for each element was prepared using certified reference materials with known concentrations of the elements of interest. The range of the calibration curve was from 0 to 100 ppm. The equation used to determine the elemental concentration in the samples was as follows:(1)C=(1−b)/m
where C is the elemental concentration (in ppm), I is the intensity of the characteristic X-ray peak for the element, b is the intercept, and m is the slope of the calibration curve. The operating conditions for the XRF analysis were as follows: voltage = 50 kV, current = 40 mA, measurement time = 60 s.

The weight percentage of oxygen (O) was measured by the commonly used difference equation as follows:(2)O=100−(C+H+N+S)

#### 2.2.2. Proximate Analysis

The proximate analysis of moisture, ash, volatile, and fixed carbon was carried out according to the standard procedure steps. Calculate the volatility based on the weight loss without the need for specific calibration curves.

Moisture determination method: Take a sample of 5–10 g and dry it in an oven at 105–100 °C for 2–4 h until the sample weight reaches a constant value.

Volatility determination method: Take a sample of 1–2 g and dry it in an oven set at 150–600 °C for 2–4 h, until the sample weight becomes constant.

Ash determination method: Take 1–2 g of sample and heat it within a range of 550 ± 25 °C in a furnace for 4 h, ensuring complete combustion of organic matter, leaving only inorganic substances.

Fixed carbon determination method: Take 1–2 g of the sample in a furnace to a high temperature within the range of 800–1000 °C, allowing the organic matter to burn off completely, leaving behind only the fixed carbon.

The equation used to determine the moisture, ash, volatile, and fixed carbon content in the samples was as follows:(3)$$X_{M}=(Xi-Xd)/Xi\times 100$$
(4)$$X_{A}=Xf/Xi\times 100$$
(5)$$X_{V}=(Xi-Xd-Xf)/Xi\times 100$$
(6)$$X_{F}=(Xi-Xd-X_{V}-Xf)/Xi\times 100$$
where X_M_ represents the moisture content of the sample; Xi represents the initial weight of the sample; and Xd represents the weight after drying of the sample; X_A_ represents the ash content of the sample; Xf represents the residue of the sample; X_V_ represents the volatility content of PS; where X_F_ represents the fixed carbon content of the sample.

The same sample element analysis and industrial molecular experiments were repeated three times, and the final results were taken three times the mean and standard deviation.

### 2.3. Thermogravimetric Analysis

The experiments were carried out using a thermogravimetric analyzer (SDTQ600, TA Instruments, New Castle, DE, USA). The PS, PP, and PS-PP (8–12 mg) were weighed and spread evenly in an alumina crucible. High-purity nitrogen (N2, 99.999%) was used with the purging rate at 25 mL·min^−1^. The heating rate was set at 10, 20, and 30 °C·min^−1^, and the experiments were performed in the 40–850 °C range.

### 2.4. Kinetic Method

In this study, all parameters of the equations were calculated using Origin (2018) and SPSS software 29 for data processing. Choose Origin and SPSS because they are widely recognized for their accuracy and reliability in data analysis and curve fitting. The experimental results of thermogravimetry at three different heating rates of 10, 20, and 30 °C·min^−1^ were fitted to reduce the uncertainty of the fitted results. According to Arrhenius’ law, the reaction kinetic equation can be obtained as follows:(7)$$\alpha \frac{d\alpha }{dT}=\frac{A}{\beta }e-\frac{Ea}{RT}\times f(\alpha )$$
where α is the conversion rate of the sample reaction; A is the frequency constant, also known as the finger front factor (min^−1^); Ea is the activation energy of the reaction (kJ·mol^−1^); β is the heating rate constant; R is the gas constant, taken as 8.314 (J·mol^−K^); T is the pyrolysis temperature (K).
(8)$$\alpha =\frac{m_{0}-m_{t}}{m_{0}-m_{f}}\times 100\%$$
where m_0_ is the starting mass of the sample pyrolysis (g); m_t_ is the mass of the sample at moment t of the reaction (g); m_f_ is the mass of solid remaining in the sample after the reaction (g).

The CR method was chosen to determine the pyrolysis mechanism function in this experiment. When n = 1, the corresponding f(α) = 1 − α f(α) = 1 − α, G(α) = −ln(1 − α); When n ≠ 1, RO(n) corresponding to the f(α) = (1 − α)n, G(α) = [(1 − (1 − α)(1 − n)))/(1 − n):(9)$$n=1,\;\;\;\;\;\;\;\ln[\frac{-\ln(1-\alpha )}{T^{2}}]=\ln(\frac{AR}{\beta E})-\frac{Ea}{RT}$$
(10)$$n\;\neq 1,\;\;\;\;\;\;\;\ln[\frac{1-(1-\alpha {)}^{1-n}}{T^{2}(1-n)}]=\ln(\frac{AR}{\beta E})-\frac{Ea}{RT}$$
where $\ln(\frac{AR}{\beta Ea})$ is a constant, a straight line is drawn in $\ln[\frac{-\ln(1-\alpha )}{T^{2}}]$ or $\ln[\frac{1-(1-\alpha {)}^{1-n}}{T^{2}(1-n)}]$ against 1/T. The reaction Ea and pre-frequency factor A is available from the slope -Ea/R and intercept $\ln(\frac{AR}{\beta Ea})$ of the line.

### 2.5. Biochar Morphology Analysis Method

The biochar obtained was collected, and its surface morphology was observed using scanning electron microscopy (SEM, Phenom XL, Eindhoven, The Netherlands) after it was naturally cooled. An acceleration voltage of 15 kV was applied during the scanning process. The surface of the biochar samples was coated with gold before the experiment to enhance the electrical conductivity and improve the quality of the images. The gold-coated samples were then scanned using SEM.

## 3. Results and Discussion

### 3.1. Characteristics of PS and PP

Table 1 shows that the half mass of PS is composed of O (51.74 wt%), which accounts for the enhanced production of oxygenated organics. The biomass’s C (42.65 wt%) and H (4.91 wt%) can be converted into valuable renewable chemical intermediates. Meanwhile, the low ash content of 12.51% in PS implied that the energy conversion would be serviceable during thermochemical conversion. PP was rich in C (84.88 wt%) and H (12.53 wt%), which can supply hydrogen to PS. The high volatile content of 99.56% in PP was conducive to reducing the reaction temperature and accelerating the reaction rate during the pyrolysis process.

### 3.2. Pyrolysis Behaviors

#### 3.2.1. Individual Pyrolysis Behavior of PS and PP

Figure 1a presents the multi-step pyrolysis process of PS with a heating rate of 10 °C·min^−1^, which could be divided into three stages. The first stage, from room temperature to 163 °C, involved dehydration and drying. The DTG curve indicated a drying temperature (T_d_) of 89 °C, which indicated the dehydration and preheating of PS [17]. The second stage was the rapid pyrolysis stage at 163–637 °C. The temperature of the maximum weight loss peak (DTG_max_) occurred around 338 °C, mainly due to the decomposition of cellulose and hemicellulose. Meanwhile, many chemical bonds were destroyed, vaporizing volatile content, such as light hydrocarbons, CO_2_, N_2_, H_2_, and CO from biomass. When the temperature exceeded 637 °C, the carbonization stage was dominated by lignin decomposition, resulting in the production of biochar and ash [4].

Figure 1b depicts the pyrolysis process of PP, which was straighter forward than that of PS and could be summarized as one-step pyrolysis. PP exhibited a flat thermogravimetric curve until 285 °C, and the DTG curve was almost a straight line [18]. During the rapid pyrolysis stage at 285–502 °C, PP depolymerized, and many polymer macromolecules decomposed into monomers or monomer derivatives. The DTG_max_ appeared near 458 °C and new volatile substances were generated by the free radical rearrangement reaction, resulting in higher volatiles and rapid weight loss [19]. As the temperature rose, the mass of the pyrolysis residue in the PP sample almost decreased to zero.

Compared to the thermal degradation rate of PS, the molecular decomposition reaction rate of PP was faster. Due to its simple structure and lack of moisture, PP required higher temperatures, which resulted in a much greater rate of weight loss in PP [20]. PP was rapidly pyrolyzed mainly by free radicals, and the produced hydrocarbons mainly originated from the thermal breakage of the C-C bond [21]. The overlapping reaction temperature range in the pyrolysis weight loss of PS and PP at a heating rate of 10 °C·min^−1^ indicated the possible co-pyrolysis interactions [22]. Thus, the research on the co-pyrolysis processes of PS and PP was further studied.

#### 3.2.2. Co-Pyrolysis Behavior of PS1-PP1

Figure 1c. shows that the co-pyrolysis behavior of PS1-PP1 consisted of two main stages, except for water loss. The first stage corresponds to the degradation of PS, while the second stage involves the cleavage of PP. At stage 1, the DTG curve of PS1-PP1 between 170 and 353 °C resembled that of PS, with the DTG_max1_ reaching 340 °C. At this time, cellulose and hemicellulose primarily contributed to decomposition, while PP had not yet started to pyrolyze [23]. With the temperature increasing, the TG curve slowed down, and PP began to soften, forming a film that enveloped the PS particles and reduced the volatile release [24]. At stage 2, the TG curve of PS1-PP1 decreased steeply at 417–635 °C and reached DTG_max2_ at 467 °C. This stage primarily involved the continuous thermal degradation of lignin and the complete degradation of PP components [25]. At 635 °C, the stage corresponded to a heating and deep polymerization stage, with significant lignin reactions, decomposition, and biochar generation [12].

The PS1-PP1 pyrolysis characteristic parameters are listed in Table 2. Compared to individual PP and PS, the addition of PP expanded the reaction temperature range of pyrolysis. These results were consistent with the findings of Chen et al. (2017) on the co-pyrolysis of lignocellulose biomass and plastic [19]. The change in pyrolysis characteristic temperature of PS1-PP1 reflected that the pyrolysis characteristics between the PS and PP were related to the interactions between the materials, further confirming the possibility of a synergistic effect during the co-pyrolysis of PS and PP [26].

### 3.3. Pyrolysis Characteristics

#### 3.3.1. Effect of Heating Rate on Pyrolysis Characteristics

The effects of the heating rate at 10, 20, and 30 °C·min^−1^ on the pyrolysis process are presented in Figure 2. With an increasing heating rate, the reaction time of PS was shortened, resulting in a whole shift to the high-temperature region of the TG and DTG curves, as observed in Figure 2a,d. Due to the accelerated rate of temperature increase, there was a delayed heat transfer, resulting in a more significant temperature difference between the internal and external regions of the biomass particles. Consequently, the dispersion of pyrolysis gases on the sample surface was impeded, leading to insufficient heating of the sample and subsequently affecting the internal pyrolysis process. Figure 2b,e shows the PP’s TG-DTG curve, exhibited a similar shift toward the high-temperature region compared to PS. Conversely, the thermal decomposition of PP was mainly focused at 300–510 °C, which required a shorter reaction time and more intense reaction. Figure 2c,f shows that the heat-loss hysteresis phenomenon of the PS1-PP1 is more pronounced compared to the individual pyrolysis. The TG-DTG curves with different heating rate analyses exhibited a consistent trend, indicating that the co-pyrolysis characteristics of the mixture were similar to pyrolysis alone [27]. In summary, the heating rate could extend the pyrolysis interval, but it did not affect the formation of volatiles. As a result, the heating rate would affect the interaction effect between PS and PP co-pyrolysis, but it did not affect the reaction mechanism.

#### 3.3.2. Effect of Blending Ratio on Pyrolysis Characteristics

The biomass blending ratio played a significant role in the pyrolysis characteristics of raw materials. The co-pyrolysis of PS-PP was more favorable than the PS alone. To determine the optimal mixing ratio and heating rate, different ratios (1:1, 1:2, 2:1) were explored at heating rates of 10, 20, and 30 °C·min^−1^, as shown in Figure 3.

As the mass of PP increased in the mixture, the pyrolysis weight loss also increased, which promoted the pyrolysis of PS at low temperatures. From the TG curve, the addition of PP changed the heat and mass transfer characteristics of PS, causing the pyrolysis T_i_ to move to the low-temperature region. Due to the significant difference in volatile yield between PP and PS, this region mainly involved the secondary pyrolysis of organic matter and the decomposition of some minerals. The DTG curve shifted to the high-temperature side, and DTG_max_ increased with the increase in PP mixing proportion. The higher volatile content of PP decreased the overall weight loss of the mixture. The presence of PP led to the formation of non-volatile polymer substances by free radical binding during thermal decomposition, and the rapid temperature increase led to the further decomposition of lignin at high temperatures to produce more volatiles [28]. The results showed that the mixing ratio affected the interaction of co-pyrolysis and thus affected the reaction mechanism.

### 3.4. Interaction between Co-Pyrolysis of PS with PP

The experiments above demonstrated a clear interaction between PS and PP during the pyrolysis process, which resulted in significant alterations in the primary reaction stages. Hence, to elucidate the co-pyrolysis synergistic mechanism, the synergistic effect was described by the difference between the experimental and calculated weight loss values (ΔW). The equation is as follows:(11)$$\Delta W={Wcal}_{exp}$$
(12)$$W_{cal}=x_{m}W_{m}+x_{p}W_{p}$$
where W_exp_ and W_cal_ denote experimental and theoretical values. x_m_ and x_p_ represent the mass percentages of PS and plastics in the sample. W_m_ and W_p_ denote the mass losses of PS and PP during individual thermal decomposition.

If ΔW was greater than 0, more volatiles were released than the expected value, indicating a synergistic effect. On the contrary, it indicates an antagonistic effect that inhibits the release of volatiles [29].

The blending ratio affects the synergistic effect of co-pyrolysis. Figure 4a shows that when the weight loss curve of PS1-PP1 co-pyrolysis at 10, 20, 30 °C·min^−1^, ΔW was greater than 0, indicating that PS1-PP1 had a synergistic effect. Before 240 °C, ΔW was set to zero, indicating a weak interaction between PS and PP. However, at 280–520 °C, ΔW was greater than 0, meaning that the co-pyrolysis process promoted the release of volatile compounds, demonstrating a synergistic effect. Under the condition of a PS: PP mass ratio of 1:1, the release of alkaline metals in the ash was more complete, and the promoting effect of PS on the pyrolysis of PP became more pronounced. After 240 °C, ΔW increased continuously, indicating that PP formed a coating on the surface of biomass particles in the molten state, thus inhibiting their volatile release [30]. As the temperature further increased, the coating effect disappeared, and the volatiles from the PS were quickly released. At this time, ΔW rapidly decreased to 0 and remained stable at 520 °C. In addition, there were differences in the ΔW curves with different heating rates, illustrating varying intensities of the synergistic effect [31]. Under the condition of a PS:PP mass ratio of 1:1, the release of alkaline metals in the ash was more complete, and the promoting effect of PS on the pyrolysis of PP became more pronounced. When the heating rate was slow, the thermal decomposition of PS was more thorough, and the formation of substances during the co-pyrolysis process of PS and PP accelerated, which helped enhance their pyrolysis reactions. In conclusion, at a heating rate of 10 °C·min^−1^, the synergistic effect of PS1-PP1 was the strongest and most favorable for promoting the release of volatiles [32].

Figure 4b depicts that in the cases of PS1-PP2, ΔW was less than 0, revealing that the excessive presence of PP inhibited the volatilization of PS below 490 °C. After 490 °C, PP melted into a molten state, which hindered the timely escape of PS volatile substances [25]. With temperatures increasing, after PP melted and macromolecular depolymerized, the bond broke and recombination continued to release heat, promoting PS pyrolysis and leading to ΔW greater than 0.

Figure 4c indicates the synergistic effect of PS2-PP1 under heating rates of 10 °C and 20 °C·min^−1^. It can be observed that the original interactions between cellulose and hemicellulose in the presence of PP were partially weakened. In contrast, the interactions between PP pyrolysis intermediates were replaced by those between cellulose and lignin pyrolysis intermediates. This statement highlights the potential synergistic interactions between PS and PP at a specific ratio. Alternatively, it is important to mention that when the heating rate was 30 °C·min^−1^, ΔW (weight change) was found to be less than 0. This observation could be explained by the fast heating rate, which caused the production of free radicals at a rate higher than the reaction rate of internal functional groups. As a result, non-volatile, high-molecular-weight substances were formed, inhibiting the reaction process [20].

To wrap up, the interaction mechanism between the PS and PP during co-pyrolysis was discussed based on the thermal behavior characteristics of the mixtures and the thermal degradation mechanisms [33]. The synergistic effect was attributed to the free radical and depolymerization reactions formed by cellulose, hemicellulose, and lignin in the binding process of PP [34]. Therefore, analyzing the co-pyrolysis processes of PS and PP using this mechanism framework was reasonable.

During the co-pyrolysis of PS-PP, PP softened and stuck to the surface of PS, hindering the heat transfer process between the PS-PP, which maintained the stability of cellulose decomposition and reduced the release of volatiles. The oxides produced from lignin degradation accelerated the breakage of the polymer chain in PS [35]. PP can serve as a hydrogen-rich resource, in which the free radical chain breakage creates hydrogen bonding, thereby facilitating the decomposition of cellulose. Additionally, it plays a role in stabilizing the primary decomposition products [36]. With the increasing temperature, PP decomposition produced hydrocarbon radical C-H bonds, which acted as reducing agents and produced more volatiles during lignin degradation at high temperatures [37]. In summary, the interaction between the oxides from cellulose and the free radicals produced from PP chain breaking created a synergistic effect. This understanding is crucial for comprehending the synergistic mechanism of PS-PP mixtures [38].

### 3.5. Kinetics Model Analysis

#### 3.5.1. Method of Kinetic Analysis

PS1-PP1 had a synergistic effect at 10, 20, and 30 °C·min^−1^. To verify the interaction of co-pyrolysis, the CR method was used to figure out the reaction kinetics of the PS, PP, and PS1-PP1. The correlation coefficients (R^2^) of the pyrolysis kinetic parameters at different heating rates remained above 0.95, confirming the accuracy and applicability of the model employed [39]. According to Table 3, PS underwent the main reaction phases at temperatures ranging from 298 to 368 °C, with corresponding Ea of 63.8, 58.9, and 64.5 kJ·mol^−1^. Similarly, for PP, the primary reaction stages occurred at temperatures between 429 and 472 °C, with Ea values of 229.2, 226.9, and 199.8 kJ·mol^−1^. Notably, both Ea and the pre-exponential factor (A) exhibited a decreasing trend as the heating rate increased. Considering that the pyrolysis process of PS1-PP1 was mainly divided into two stages, the Ea calculations were performed for each stage. Stage 1 corresponded to temperatures ranging from 290 to 253 °C, with Ea values of 45.6, 42.7, and 45.8 kJ·mol^−1^. Stage 2 occurred between 417 and 482 °C, with Ea of 93.8, 73.6, and 86.5 kJ·mol^−1^.

Compared to the pyrolysis of PS and PP alone with that of PS1-PP1, Ea at stage 1 was lower than that during the pyrolysis of PS. This indicated that the existence of PP had no significant influence on the pyrolysis of PS before 350 °C, as PP had not undergone pyrolysis yet. In other cases, Ea was significantly lower in the second stage than in the PP stage, indicating that the addition of PP reduced the energy required for pyrolysis, primarily due to the synergistic effect between PS and PP. This finding aligns with the results obtained in Section 3.3 of the study. In general, the decrease of Ea in PS1-PP1 indicated that there was a synergistic effect between PS and PP, so it is necessary to explore the synergistic mechanism further.

#### 3.5.2. Kinetic Mechanism Functions

To further determine the kinetic parameters and reaction mechanism of the co-pyrolysis, six sets of values were taken for the number of reaction orders, *n* = 0.5, 1, 1.5, 2, 2.5, and 3 at 10 °C·min^−1^ to bring the experimental data into Equations (9) and (10) for the least-squares fitting, and the linear fitting results are shown in Table 4.

In the case of PS, R^2^ increases first and then decreases with the increase of *n*. When *n* = 1.5, the linear correlation coefficient is the largest (R^2^ = 0.984), Ea = 78.65 kJ·mol^−1^, and the linear fitting equation is Y = −9.46X + 2.68. It can be deduced that the pyrolysis kinetic expression of a peanut shell is as follows:(13)$$\frac{d\alpha }{dT}=137.8e^{-\frac{94603}{T}}\cdot (1-\alpha {)}^{1.5}$$

In the case of PP, different from PS, the R^2^ decreases continuously with the increase of *n*. When *n* = 0.5, the linear correlation coefficient is the largest (R^2^ = 0.999), the linear fitting equation is Y = −21.31X + 15.81, the Ea of PP pyrolysis is 177.19 kJ·mol^−1^, and the A = 1.57 × 10^9^. Studies showed that PP required more Ea than PS for complete pyrolysis, and plastic pyrolysis required more energy than biomass pyrolysis. It can be deduced that the pyrolysis kinetic expression of PP is as follows:(14)$$\frac{d\alpha }{dT}=1.57\times 10^{8}e^{-\frac{21311.84}{T}}\cdot (1-\alpha {)}^{0.5}$$

In the case of PS1-PP1, the R^2^ of the mechanism function corresponding to the two stages decreased as the number of n rose. The stage 1 correlation coefficient is maximum (R^2^ = 0.997) at *n* = 0.5, and the linear fit equation is Y = −5.13X − 5.75935, Ea = 42.63 kJ·mol^−1^, A = 0.16. The stage 2 correlation coefficient was the largest (R^2^ = 0.946), and the linear fitting equation was Y = −8.29X − 2.05, Ea = 68.93 kJ·mol^−1^ and A = 10.69. Accordingly, the multistage reaction mechanism function model can react to the mechanism of the primary reaction process of co-pyrolysis. It can be deduced that the pyrolysis kinetic expression of PS1-PP1 is as follows:(15)$$Stage\;1:\;\;\;\;\;\;\;\frac{d\alpha }{dT}=0.017e^{-\frac{5127.86}{T}}\cdot (1-\alpha {)}^{0.5}$$
(16)$$Stage\;2:\;\;\;\;\;\;\;\frac{d\alpha }{dT}=1.07e^{-\frac{8290.34}{T}}\cdot (1-\alpha {)}^{0.5}$$

Overall, the Ea of the PS1-PP1 co-pyrolysis was significantly reduced compared to the Ea of the pyrolysis alone, indicating that the mixed pyrolysis favored the reaction. Hence, a significant synergism occurred between PP and PS. This also agreed with Huang et al. (2020), who used the CR method to explore the reaction mechanism of co-pyrolysis of polyolefin plastics and biomass [22].

The decrease in Ea during co-pyrolysis could be attributed to the partial decomposition of cellulose into furans during the pyrolysis of PS. These furans formed aromatic compounds (e.g., benzenes) by undergoing Diels–Alder synthesis and deoxygenation reactions with substances such as ethylene and propylene generated from the depolymerized PP during pyrolysis [40]. The transfer of intermolecular radicals in DA and deoxygenation reactions could reduce the intermolecular stability of the sample and, thus, the Ea of the reaction [41]. Additionally, the diffusion resistance of biomass decreased during co-pyrolysis, facilitating the movement of volatiles and driving the pyrolysis reaction forward. The co-pyrolysis of PS and PP exhibited synergistic effects, as evidenced by the reduced Ea and the interactions between furans from PS and depolymerized PP. The diffusion of volatiles also contributes to the overall pyrolysis reaction.

### 3.6. SEM Analysis

The properties of biochar obtained from co-pyrolysis differ from those of single-pyrolysis carbon, highlighting the importance of studying co-pyrolysis biochar to understand the interaction between co-pyrolysis particles and between volatiles and particles. Biochar samples were analyzed by the scanning electron microscopy (SEM) scanning technique to reveal their structural properties and morphology and to understand their structural changes during pyrolysis further. In addition, the thermogravimetric analysis showed that PP had less residue after degradation, so the PS1-PP1 biochar consisted mainly of char formed by PS degradation.

As shown in Figure 5a–c, the surface morphology of PS biochar exhibited irregular layered loose material with irregular and rough pore structure. The reason was that the precipitation of volatile components and the plastic deformation of biochar at high temperatures hindered the formation of pores and led to structural distortion [42]. Crystalline particles were also observed on the surface of PS char, likely resulting from the melting of low-melting-point ash and condensation on the surface of the char [43]. Figure 5d–f shows that after adding PP to the PS, the number of pores in the pyrolytic carbon decreases and the diameter increases, indicating a looser structure [44]. Deep polymerization of PP promoted the volatile release in PS [45]. In addition, the heating rate also influenced the layer structure to become thinner, the surface roughness increased, and micro-pores were formed, but some of the pores were blocked. When the temperature rose to 10 °C·min^−1^, the lamellar structure was reorganized, the blocked holes were opened, and the accumulation of lamellar structures of different sizes formed irregular holes. Mixtures have minor ash at PS1-PP1, indicating the complete reaction and the synergistic effect was most potent from the point of view of pyrolysis products. This was the same as the conclusion of the synergy analysis in the Section 3.3 co-pyrolysis process. This indicated that the specific relationship between coke surface morphology and the PS1-PP1 ratio will be further studied in the future.

## 4. Conclusions

Co-pyrolysis expanded the pyrolysis temperature range, and the addition of PP reduced the Ea required for biomass pyrolysis, resulting in increased volatiles. The heating rate affects the peak temperature but does not change the reaction mechanism of the material. The mixing ratio has a significant effect on the co-pyrolysis behavior of biomass and PP and will change the interaction mechanism. It is worth noting that when the mass ratio of PS to PP is 1:1, positive synergies are observed at different heating rates, and the synergies are most potent at a 10 °C·min^−1^ heating rate. The CR model has good fitting performance in the separate pyrolysis kinetics calculations of PS, PP, and PS1-PP1 and reduces the Ea. The multi-stage reaction mechanism function model was used to get the optimal mechanism function fit, in which the pyrolysis mechanism was the most accurate when *n* = 0.5. The structural properties and morphology of biochar obtained from co-pyrolysis were changed by SEM analysis, indicating that co-pyrolysis affected carbon formation. This synergistic effect is attributed to the interaction between cellulose, hemicellulose, and lignin in the biomass and the free radicals generated by PP chain breaking. These findings provide valuable insights into co-pyrolysis processes and the potential use of waste plastics and biomass in sustainable energy production.

## Figures and Tables

**Figure 1 foods-13-01191-f001:**
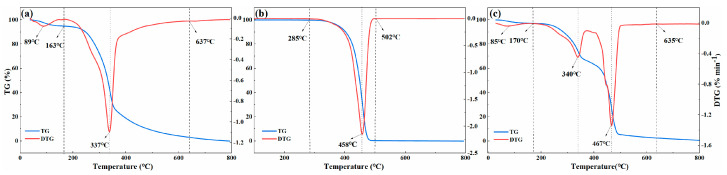
TG and DTG curves of PS, PP, and PS1-PP1 at 10 °C min^−1^ heating rate. (**a**) TG-DTG curve of PS pyrolysis; (**b**) TG-DTG curve of PP pyrolysis; (**c**) TG -DTGcurve of PS1-PP1 pyrolysis.

**Figure 2 foods-13-01191-f002:**
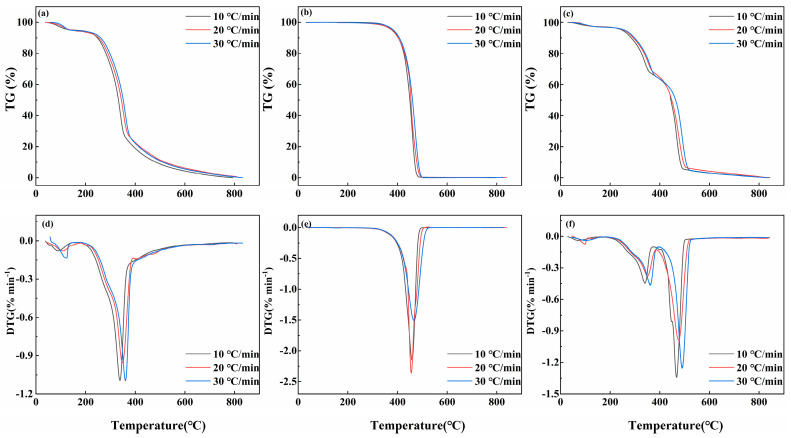
TG−DTG curves of PS and PP pyrolysis at three different heating rates. (**a**) TG curve of PS pyrolysis; (**b**) TG curve of PP pyrolysis; (**c**) TG curve of PS1-PP1 pyrolysis; (**d**) DTG curve of PS pyrolysis; (**e**) DTG curve of PP pyrolysis; (**f**) DTG curve of PS1-PP1 pyrolysis.

**Figure 3 foods-13-01191-f003:**
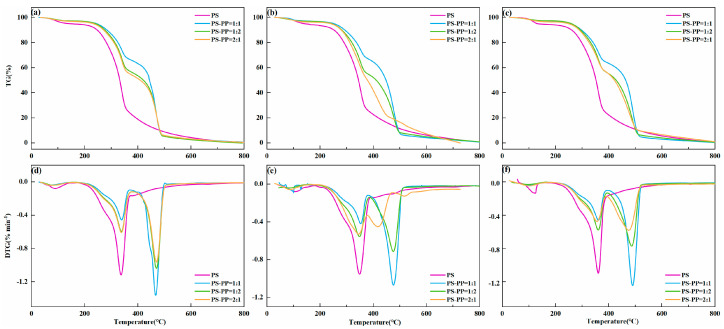
TG and DTG curves of the PS−PP at mass ratios of 1:1, 1:2, and 2:1 and under 10, 20, and 30 °C·min^−1^. (**a**) TG curve of PS−PP pyrolysis at 10 °C⋅min^−1^ heating rate; (**b**) TG curve of PS−PP pyrolysis at 20 °C⋅min^−1^ heating rate; (**c**) TG curve of PS−PP pyrolysis at 30 °C⋅min^−1^ heating rate; (**d**) DTG curve of PS−PP pyrolysis at 10 °C⋅min^−1^ heating rate; (**e**) DTG curve of PS−PP pyrolysis at 20 °C⋅min^−1^ heating rate; (**f**) DTG curve of PS−PP pyrolysis at 30 °C⋅min^−1^ heating rate.

**Figure 4 foods-13-01191-f004:**
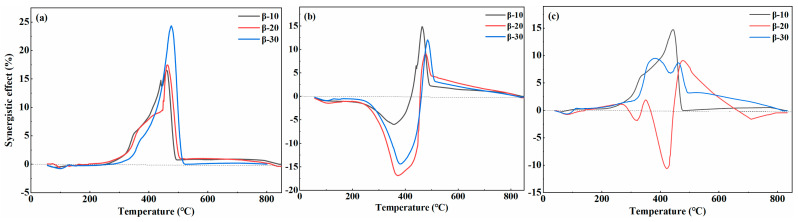
TG curves comparison between the experimental and calculated value from the blends 1:1, 1:2, and 2:1 of under 10, 20, and 30 °C·min^−1^. (**a**) ΔW curves for PS1−PP1 at 10, 20, 30 °C⋅min^−1^; (**b**) ΔW curves for PS1−PP2 at 10, 20, 30 °C⋅min^−1^; (**c**) ΔW curves for PS2−PP1 at 10, 20, 30 °C⋅min^−1^.

**Figure 5 foods-13-01191-f005:**
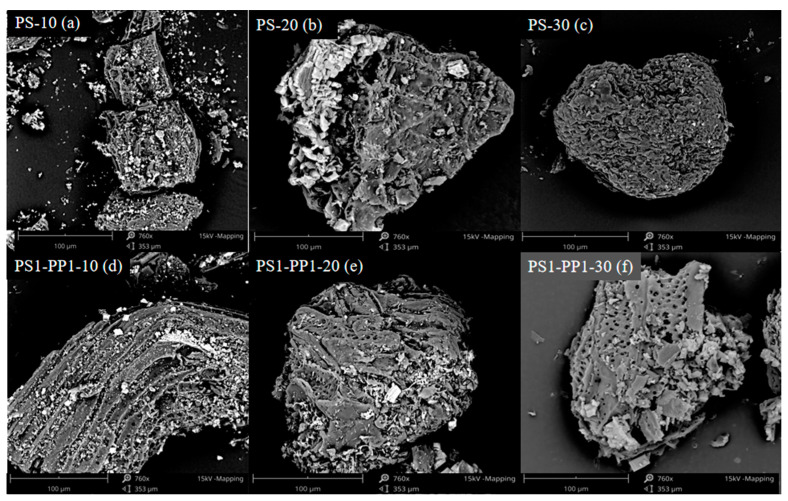
SEM analyses of the PS and PS1-PP1 biochar samples prepared under 10, 20, and 30 °C·min^−1^. (**a**) SEM analysis of PS at 10 °C·min^−1^ heating rate; (**b**) SEM analysis of PS at 20 °C·min^−1^ heating rate; (**c**) SEM analysis of PS at 30 °C·min^−1^ heating rate; (**d**) SEM analysis of PS1-PP1 at 10 °C·min^−1^ heating rate; (**e**) SEM analysis of PS1-PP1 at 20 °C·min^−1^ heating rate; (**f**) SEM analysis of PS1-PP1 at 30 °C·min^−1^ heating rate;.

**Table 1 foods-13-01191-t001:** Elemental analysis of biomass samples, industrial analysis.

Sample	Elemental Analysis (wt%)					Proximate Analysis (wt%)			
	C	H	O	N	S	Moisture	Volatile	Fixed Carbon	Ash
PS	41.56 ± 0.88	4.87 ± 0.34	50.47 ± 0.45	0.57 ± 0.12	0.11 ± 0.04	5.78 ± 0.21	62.07 ± 1.23	20.78 ± 0.32	11.57 ± 0.03
PP	83.76 ± 0.85	11.35 ± 0.88	20.56 ± 0.23	0	0	0.68 ± 0.08	99.45 ± 0.75	0.02 ± 0.0.04	0.01 ± 0.036

Note: Results are expressed as mean ± standard deviation (sd) and three repetitions to take an average value of experimental parameters.

**Table 2 foods-13-01191-t002:** Main characteristic values of the PS, PP, and PS1-PP1 at 10 °C·min^−1^.

Sample	β	T_d_	T_i_	T_f_	T_max_	DTG_max_	Residue (%)
	(°C·min^−1^)	(°C)	(°C)	(°C)	(°C)	(%·min^−1^)	
PS	10	88 ± 0.8	163 ± 1.7	635 ± 0.4	335 ± 2.1	−1.09 ± 0.33	0.03 ± 0.07
PP	10	/	285 ± 1.5	498 ± 0.78	457 ± 3.54	−2.14 ± 0.12	0.00 ± 0.06
PS1-PP1	10	85 ± 0.63	170 ± 0.41	633 ± 2.7	340 ± 3.7/467 ± 2.36	−0.44 ± 0.08/−1.33 ± 0.10	0.57 ± 0.25

Note: β: heating rate; T_d_: drying temperature; T_i_: initial temperature; T_f_: final temperature; T_max_: maximum temperature; DTG_max_ maximum temperature from the DTG curves. Results are expressed as mean ± standard deviation (sd) and three repetitions to take an average value of experimental parameters.

**Table 3 foods-13-01191-t003:** Kinetic parameters of co-pyrolysis of PS, PP, and PS1-PP1 at different heating rates.

Sample	β	Temperature Range	Ea	A	R^2^
	(°C·min^−1^)	(°C)	(kJ·mol^−1^)	(min^−1^)	
PS	10	298–368	63.82	792.64	0.984
	20	305–382	58.97	433.28	0.986
	30	315–387	64.54	1785.058	0.988
PP	10	429–472	229.27	14,804.09 × 10^9^	0.992
	20	435–476	226.93	14,801.35 × 10^9^	0.990
	30	432–493	199.83	176.97 × 10^9^	0.989
PS1-PP1	10	290–353	45.62	5.57	0.996
		417–482	93.82	19,334.38	0.924
	20	297–365	42.76	4.84	0.998
		412–495	73.60	969.35	0.924
	30	308–375	45.85	13.29	0.996
		435–507	86.49	10,336.71	0.900

**Table 4 foods-13-01191-t004:** Kinetic fitting parameters at each reaction level for PS, PP, and PS1-PP1.

Sample		*n*	R^2^	Y	Ea
					kJ mol^−1^
PS		0.5	0. 981	−6.126X − 3.23	50.93
		1	0.984	−7.68X − 0.48	63.82
		1.5	0.984	−9.46X + 2.68	78.65
		2	0.983	−11.47X + 6.23	95.39
		2.5	0.980	−13.70X + 10.15	113.91
		3	0.977	−16.12X + 14.39	134.02
PP		0.5	0.999	−21.31X + 15.81	177.19
		1	0.992	−27.58X + 24.71	229.27
		1.	0.974	−35.31X + 35.66	293.54
		2	0.951	−44.41X + 48.54	369.22
		2.5	0.929	−54.66X + 63.02	454.41
		3	0.912	−65.79X + 78.75	546.96
PS1-PP1	Stage1	0.5	0.997	−5.13X − 5.76	42.63
		1	0.996	−5.49X − 5.10	45.62
		1.5	0.995	−5.86X − 4.41	48.73
		2	0.994	−6.25X − 3.70	51.96
		2.5	0.993	−6.65X − 2.96	55.31
		3	0.992	−7.07X − 2.20	58.78
	Stage2	0.5	0.946	−8.29X − 2.049	68.93
		1	0.924	−11.28X + 2.33	93.81
		1.5	0.901	−14.940X + 7.66	124.21
		2	0.880	−19.22X + 13.876	159.83
		2.5	0.863	−24.06X + 20.88	200.01
		3	0.851	−29.33X + 28.51	243.85

Note: *n*: reaction order; R^2^: correlation coefficient; Y: fitting equation.

## Data Availability

The original contributions presented in the study are included in the article, further inquiries can be directed to the corresponding author.

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
