# Peer review of "Synergistic Effects and Kinetic Analysis in Co-Pyrolysis of Peanut Shells and Polypropylene"

_foods, 2024, doi:10.3390/foods13081191_

Round 1

Reviewer 1 Report

Comments and Suggestions for Authors

The article focuses on the co-pyrolysis of peanut shells (PS) and polypropylene (PP) as an innovative approach for managing plastic waste and enhancing energy efficiency. Through thermogravimetric analysis at various heating rates, it was discovered that the co-pyrolysis of PS and PP occurs in two main stages: the first stage (170–400°C) primarily involves the decomposition of PS, while the second stage (400-520°C) is characterized by the combined thermal degradation of both PS and PP, with PP degradation being more significant. The study observed thermogravimetric hysteresis with increasing heating rates and found that co-pyrolysis lowers the activation energy required for pyrolysis, particularly in the second stage, maintaining a high correlation coefficient (R^2 > 0.95). A multi-level reaction mechanism function model effectively unveiled the co-pyrolysis mechanism, demonstrating that PS is a high-quality biomass for co-pyrolysis with PP. Synergistic effects were observed in all mixtures at a 1:1 mixing ratio of PS to PP, indicating efficient waste utilization and optimized energy consumption. This research is pivotal in understanding the interaction mechanisms of mixed samples in the co-pyrolysis process, presenting a significant stride towards sustainable waste management and energy recovery. Idea sounds good but there are some minor points that should be considered:

Line 2: Please remove the words that used in the title from keywords and suggest different ones.

Tables: Tables look s o big. Please arrange the fonts to fit text.

Line 14: Give a few sentences relating to your work at first.

Statistical analysis: Why did not you calculate statistical values? Did you do only one time your experiments?

Did not you use any software for calculating parameters of the equations?

Comments on the Quality of English Language

Minor editing of English language required.

Author Response

Responses to Reviewers

Manuscript Number: foods-2934480

Answers to Editor and Reviewers.

Reviewer #1

The article focuses on the co-pyrolysis of peanut shells (PS) and polypropylene (PP) as an innovative approach for managing plastic waste and enhancing energy efficiency. Through thermogravimetric analysis at various heating rates, it was discovered that the co-pyrolysis of PS and PP occurs in two main stages: the first stage (170–400°C) primarily involves the decomposition of PS, while the second stage (400-520°C) is characterized by the combined thermal degradation of both PS and PP, with PP degradation being more significant. The study observed thermogravimetric hysteresis with increasing heating rates and found that co-pyrolysis lowers the activation energy required for pyrolysis, particularly in the second stage, maintaining a high correlation coefficient (R^2 > 0.95). A multi-level reaction mechanism function model effectively unveiled the co-pyrolysis mechanism, demonstrating that PS is a high-quality biomass for co-pyrolysis with PP. Synergistic effects were observed in all mixtures at a 1:1 mixing ratio of PS to PP, indicating efficient waste utilization and optimized energy consumption. This research is pivotal in understanding the interaction mechanisms of mixed samples in the co-pyrolysis process, presenting a significant stride towards sustainable waste management and energy recovery. Idea sounds good but there are some minor points that should be considered:

Response: Thank you for taking the time to review the manuscript and providing valuable feedback. I greatly appreciate your input. Based on your suggestions, I have made the necessary revisions, which have been clearly marked in red in the main text.Thank you again for your diligent review and your valuable contributions. I have carefully considered your comments and incorporated them into the revised version. Your expertise and insights have greatly improved the quality of the manuscript. Please let me know if there are any additional changes or revisions that you would recommend.     I look forward to hearing your thoughts on the revised manuscript.

  1. Line 2: Please remove the words that used in the title from keywords and suggest different ones.

Response: Thanks for the suggestion. The keywords that repeat the title have been removed and changed to “Co-pyrolysis prorcess; Thermogravimetry; heating rate; Activation energy; Synergistic mechanism” (Line 30)

  1. Tables: Tables look so big. Please arrange the fonts to fit text.

Response: We have adjusted the size of each table according to your suggestions to ensure that the text content can be displayed fully and aesthetically. To maintain integrity, we adjusted the text content in Table to a 9-point font. This adjustment ensures the table content is clear and easy to read. (Lines 146,186, 357, and 385)

  1. Line 14: Give a few sentences relating to your work at first.

Response: Thank you for your advice. To ensure consistency in your summary, add a sentence to the first line. “The impact of COVID-19 has boosted the growth in the takeaway and medical industries but has also generated a large amount of plastic waste. Peanut shells (PS) are produced in large quanti-ties with challenging to recycle in China.”(Lines 14-16)

  1. 4. Statistical analysis: Why did not you calculate statistical values?

Response: I apologize for the oversight in not including statistical values in Tables 1 and 2. I will revise this by promptly calculating and adding the necessary statistical values. In addition, we consider that Tables 3 and 4 are not suitable for statistical analysis. The reasons are as follows: The data in Table 3 and Table 4 are based on the thermogravimetric analysis of PS and PP, and the co-pyrolysis behavior is further illustrated through first-order kinetic model calculation and equation fitting. Our focus is on describing macroscopic phenomena in the main pyrolysis weight loss range to provide a basis for answering the question of co-pyrolysis interactions. The results provided a theoretical basis for the recycling and energy utilization of PS and PP.

  1. Did you do only one time your experiments?

Response: Thank you for your feedback. I would like to clarify that our experiments were conducted three times to ensure the reliability and robustness of our results. We appreciate your attention to this matter and are committed to providing accurate and rigorous scientific research. It has been described in Experiments and Methods 2.1 and 2.2. (Lines 107-109)

  1. Did not you use any software for calculating parameters of the equations?

Response: Thank you for your inquiry. In our study, all parameters of the equations were calculated using Origin and SPSS software for data processing. We chose Origin and SPSS because it is widely recognized for its accuracy and reliability in data analysis and curve fitting. We ensured that the software was properly calibrated and that the most appropriate fitting models were selected for each equation. The parameters obtained from Origin and SPSS were then used in our analysis and discussion. We believe that using Origin and SPSS software enhanced the accuracy and reliability of our results “In this study, all parameters of the equations were calculated using Origin and SPSS software for data processing. Chose Origin and SPSS because it is widely recognized for its accuracy and reliability in data analysis and curve fitting.”(Lines 117-119)

Reviewer 2 Report

Comments and Suggestions for Authors

MANUSCRIPT: 2934480

TITLE: Synergistic Effects and Kinetic Analysis in the Co-Pyrolysis of Peanut Shells and Polypropylene

 The manuscript 2934480 “Synergistic Effects and Kinetic Analysis in the Co-Pyrolysis of Peanut Shells and Polypropylene”, presents a very good work on evaluating the application and use of waste or food waste in a green technology for reducing or eliminating polluting products.

It is a manuscript that reveals a scientific work. The methodologies are appropriate to the objectives of the work.

This work is well structured, well planned and the research is competently carried out.

Methodology used was adequate and appropriate to the objectives of the work but not fully described.

The results were subject to an appropriate statistical analysis.

These works are very innovative and important in terms of finding sustainable alternatives to mitigate the environmental impact that food waste and packaging can have on the environment.

The literature is well cited and most of the papers cited (> 75%) date back to the last five years.

Conclusions are presented according to the results obtained.

I congratulate the authors for this work of high interest to the scientific community.

However, I have some comments to be clarified and resolved in the current manuscript:

1. The manuscript presents many abbreviations without being written in full the first time they appear in the text of the manuscript (e.g. Line 58 LDPE) It is recommended to carefully review the text and present a list of abbreviations as many abbreviations are used without being defined in full the first time they are used.

2. The manuscript presents results in table 1 on elemental analysis and proximate analysis and these methodologies are not described in section 2. Materials and Methods. Please, to allow the replication of the study, describe in detail all the methods for each parameter determined, including in all of them the complete description of the methodologies. It is therefore recommended that they be indicated unequivocally in all methods, namely: range of calibration curve, equation used to determine parameters and operating conditions used in some instrumental analysis techniques.

3. Section 2. Materials Methods – Please carefully review this section and in all subsections consider adding in all instruments used, model, producer, and its location (Instrument model, Producer, City, State Abbr., Country). Proceed in the same way for all instruments used.

4. Table 1 – Please in table 1 the results of determined parameters should be expressed as mean ± standard deviation (S.D.).

5. Table 1 - Please indicate in footnote: The results are expressed as mean ± standard deviation (S.D.) and the (n) number of replicates in each experimental parameter determined.

Author Response

Responses to Reviewers

Manuscript Number: foods-2934480

Answers to Editor and Reviewers.

Reviewer #2

The manuscript 2934480 “Synergistic Effects and Kinetic Analysis in the Co-Pyrolysis of Peanut Shells and Polypropylene”, presents a very good work on evaluating the application and use of waste or food waste in a green technology for reducing or eliminating polluting products.

It is a manuscript that reveals a scientific work. The methodologies are appropriate to the objectives of the work.

This work is well structured, well planned and the research is competently carried out.

Methodology used was adequate and appropriate to the objectives of the work but not fully described.

The results were subject to an appropriate statistical analysis.

These works are very innovative and important in terms of finding sustainable alternatives to mitigate the environmental impact that food waste and packaging can have on the environment.

The literature is well cited and most of the papers cited (> 75%) date back to the last five years.

Conclusions are presented according to the results obtained.

I congratulate the authors for this work of high interest to the scientific community.

However, I have some comments to be clarified and resolved in the current manuscript:

Response: Thank you for taking the time to review the manuscript and providing valuable feedback. I greatly appreciate your input. Based on your suggestions, I have made the necessary revisions, which have been clearly marked in red in the main text.Thank you again for your diligent review and your valuable contributions. I have carefully considered your comments and incorporated them into the revised version. Your expertise and insights have greatly improved the quality of the manuscript. Please let me know if there are any additional changes or revisions that you would recommend.     I look forward to hearing your thoughts on the revised manuscript.

  1. The manuscript presents many abbreviations without being written in full the first time they appear in the text of the manuscript (e.g. Line 58 LDPE) It is recommended to carefully review the text and present a list of abbreviations as many abbreviations are used without being defined in full the first time they are used.

Response: Thank you very much for pointing out the problem about the use of abbreviations. At your suggestion, we have replaced "LDPE" to "low density polyethylene" at line 61 and "thermal TGA" to "thermal gravimetric analyzer" at line 77 to ensure that the initial abbreviation can be fully interpreted. In addition, we have added an abbreviations section that lists all the abbreviations used in the text and their full names for easier reference by readers. Thank you for your valuable comments. We will carefully review the manuscript and make changes accordingly to improve the readability and quality of the paper. Abbreviations have been appended below the Acknowledgmentst for ease of reading. (Line 61, 77 and 445) (see below)

Abbreviation

PS

peanut shells

Y

fitting equation

PP

polypropylene

m0

the initial mass (g) of the sample

PS-PP

mixture of PS and PP

mt

the sample mass (g) at the time point of the reaction t

PS1-PP1

Mixture ratio of 1:1PS and PP

mf

the mass of solid remaining in the sample after the reaction (g)

PS1-PP2

Mixture ratio of 1:2PS and PP

SEM

sanning electron microscopy

PS2-PP1

Mixture ratio of 2:1PS and PP

Td

the drying temperature

R2

correlation coefficient

DTGmax

maximum temperature from the DTG curves

Ea

activation energy

Ti

initial temperature

CR

Coats-Redfern

Tf

final temperature

α

mass conversion rate

Tmax

maximum temperature

A

pre-exponential factor

Wexp

experimental values

β

 the heating rate constant

Wcal

theoretical values

R

universal gas constant (8.314 J/(mol·K))

xm

the mass percentages of PS

T

the pyrolysis temperature (K)

xp

the mass percentages of plastics

n

reaction order

Wm

the mass losses of PS during thermal decomposition

f(α)

reaction mechanism function

Wp

the mass losses of PP during thermal decomposition

  1. The manuscript presents results in table 1 on elemental analysis and proximate analysis and these methodologies are not described in section 2. Materials and Methods. Please, to allow the replication of the study, describe in detail all the methods for each parameter determined, including in all of them the complete description of the methodologies. It is therefore recommended that they be indicated unequivocally in all methods, namely: range of calibration curve, equation used to determine parameters and operating conditions used in some instrumental analysis techniques.

Response: Thank you very much for your attention. To ensure the reproducibility of our results, we have updated the method descriptions of all parameters in the paper, including a detailed description of the experimental methods and operating conditions. In particular, in Section 2.1, we have added method descriptions for elemental analysis and approximate analysis, and determined parametric equations. In all sections and sub-sections, we indicate exactly the operating conditions of the range of calibration curves, equations and instrumental analysis techniques used to determine the parameters. I hope these updates meet your requirements. The specific contents are as follows. (Lines 92-109)

2.2 sample analysis

The main elemental compositions (C, H, N, and S) of the samples were determined by an elemental analyzer (Vario EL cube, Elementar, Frankfurt, Germany). Elemental analysis was performed using the X-Ray Fluorescence (XRF) technique. The calibration curve for each element was prepared using certified reference materials with known concentrations of the elements of interest. The range of the calibration curve was from 0 to 100 ppm. The equation used to determine the elemental concentration in the samples was:

(1)

where C is the elemental concentration (in ppm), I is the intensity of the characteristic X-ray peak for the element, b is the intercept, and m is the slope of the calibration curve.   The operating conditions for the XRF analysis were as follows: voltage = 50 kV, current = 40 mA, measurement time = 60 seconds.

The weight percentage of oxygen ( O ) was measured by the commonly used difference equation as follows:

(2)

In addition, the proximate analysis of moisture, volatile, fixed carbon, ash, were evaluated according to the GB / T2677.2-1993 standard method. The determination was carried out according to the standard procedure steps. The same sample element analysis and industrial molecular experiments were repeated three times, and the final results were taken three times the mean and standard deviation. “

3) Section 2. Materials Methods – Please carefully review this section and in all subsections consider adding in all instruments used, model, producer, and its location (Instrument model, Producer, City, State Abbr., Country). Proceed in the same way for all instruments used.

Response: Thank you for the reminder. We have carefully reviewed the section and have added the information regarding the instruments used, their models, producers, and their respective locations. Add the multi-functional pulveriser (XT-A400, Xintao, Yongkang city, China) of 2.1. Add the city Frankfurt. Add the city Eindhoven in section 2.4. (Lines 88,94 and 138)

  1. Table 1. Please in table 1 the results of determined parameters should be expressed as mean ± standard deviation (S.D.).

Response: Thank you for your advice. We have now updated Table 1 to express the results of determined parameters as mean ± standard deviation (S.D.).(Line 146 )

  1. Table 1 Please indicate in footnote: The results are expressed as mean ± standard deviation (S.D.) and the (n) number of replicates in each experimental parameter determined.

Response: Thank you for your attention to detail. I have added the relevant information in the footnote according to your suggestion. (Line147)

Round 2

Reviewer 2 Report

Comments and Suggestions for Authors

MANUSCRIPT: 2934480

TITLE: Synergistic Effects and Kinetic Analysis in the Co-Pyrolysis of Peanut Shells and Polypropylene

In the revised manuscript 2934480 “Synergistic Effects and Kinetic Analysis in the Co-Pyrolysis of Peanut Shells and Polypropylene”, presents the authors present the new manuscript reformulated according to almost all the reviewers' recommendations.

Regarding the manuscript presented, I congratulate the authors for their effort in considering almost all reviewers' suggestions and for the valuable work presented. However, the methods used in the proximal analysis should be presented in more detail.

I have some comments to authors consider in the manuscript:

1. The manuscript presents results in table 1 on proximate analysis and these methodologies are not described in section 2.

2. Please include in the Materials and Methods section to allow the replication of the study a detailed and complete description of the methods used for the determination of moisture, volatile, fixed carbon, and ash.

3. An in-detail description of the proximate analysis methods must include sample preparation indicating the amount of sample used, the equations used or the range of calibration curve to determine parameters and operating conditions used if instrumental analysis techniques are used.

Author Response

Responses to Reviewers

Manuscript Number: foods-2934480

Answers to Editor and Reviewers.

Reviewer #2

In the revised manuscript 2934480 “Synergistic Effects and Kinetic Analysis in the Co-Pyrolysis of Peanut Shells and Polypropylene”, presents the authors present the new manuscript reformulated according to almost all the reviewers' recommendations.

Regarding the manuscript presented, I congratulate the authors for their effort in considering almost all reviewers' suggestions and for the valuable work presented. However, the methods used in the proximal analysis should be presented in more detail.

I have some comments to authors consider in the manuscript:

Response: I would like to express my sincere gratitude for taking the time to review my manuscript and providing valuable feedback. I have carefully considered your comments and made the necessary revisions as indicated by the blue markings in the main text.

Please see the response in the attached file.

Once again, thank you for your assistance in improving the quality of my work.
